# Cultural Significance of Beetles in Sub-Saharan Africa

**DOI:** 10.3390/insects12040368

**Published:** 2021-04-20

**Authors:** Arnold van Huis

**Affiliations:** Laboratory of Entomology, Wageningen University & Research, P.O. Box 16, 6700 AA Wageningen, The Netherlands; arnold.vanhuis@wur.nl

**Keywords:** beetles, Coleoptera, insects as food, medicine, toys, superstition, stories, proverbs

## Abstract

**Simple Summary:**

The article explains how beetles are used, perceived, and experienced in daily life across sub-Saharan Africa. More than 300 people from 27 countries were interviewed. The results were compared with findings in the scientific literature. Many beetle species are eaten, both as adults and larvae. Young girls use water beetles to stimulate breast growth. Blister beetles contain a substance that is used as an aphrodisiac and for treating venereal disease. Throughout Africa children play with beetles by letting them fly at the end of a string. In the Sahel dung beetles are predominant, and there are stories relating to how hard they work to remove dung. The magical glowing fireflies are associated with ghosts and witches. Many beetle groups are brightly and attractively coloured, and are therefore used in ceremonies, as ornaments, and in commerce. However, urbanisation is resulting in the rapid disappearance of this type of indigenous knowledge.

**Abstract:**

An overview is given of how beetles are utilised, perceived, and experienced in daily life across sub-Saharan Africa. More than 300 people from 27 countries were interviewed and the results were compared with literature findings. Both the adults and larvae of many beetle species are eaten, mainly from the families Curculionidae, Buprestidae, Cerambycidae, Dytiscidae and Scarabaeidae. Some beetle species are used for medicinal purposes. The use of breast-shaped water beetles by adolescent girls to stimulate breast growth could be due to superstition or the effect of the defensive secretion containing steroid hormones. Blister beetles contain cantharidin, which influences the urinary tract, and is therefore used as an aphrodisiac and for treating venereal disease. Throughout Africa children play with beetles by letting them fly on a string. In the Sahel, the hard work of dung beetles is an inspiration for stories. Fireflies are generally associated with ghosts and witches, likely because the glowing of the beetles is interpreted as magical. Many beetle groups are brightly and attractively coloured and are therefore used in ceremonies and commerce. This type of indigenous knowledge, revealed in stories provided by older people, is rapidly disappearing due to urbanisation.

## 1. Introduction

Beetles belong to the order of the Coleoptera, with 376,363 described species in 43,260 genera; it is the largest order on earth, containing 34.2% of all insect species [1]. These holometabolous insects have a great anatomical and behavioural diversity. Most beetles can fly. When not in use, their flight wings are concealed below protective elytra, permitting them to use diverse resources and engage in a broad range of activities that otherwise would be restricted to either winged or wingless species [2]. Most beetles are herbivores, fungivores and carnivores, both in adult and larval stages. 

There are two suborders, the Adephaga and the Polyphaga. From the Adephaga the most frequently mentioned families in the interviews are the Dytiscidae, Gyrinidae, and Carabidae. From the Polyphaga, the largest suborder considering all described species, the families most frequently mentioned are Buprestidae, Cerambycidae, Chrysomelidae, Curculionidae, Lampyridae, Meloidae, Scarabaeidae and Tenebrionidae. We will briefly describe the different families and use the Global Biodiversity Information Facility website [1] to indicate between parentheses the number of species and genera occurring worldwide (spp.; genera).

Water beetles and ground beetles belong to the Adephaga. Dytiscidae (4511; 233) are predaceous diving beetles, and Gyrinidae (701; 34) are whirligig beetles; both are found in standing water. Gyrinidae usually swim on the surface of the water if undisturbed, and underwater when threatened. They get their common name from their habit of swimming rapidly in circles. Carabidae (40,616; 2659) (ground beetles) are often found under the bark of trees, under logs, or among rocks or sand by the edge of ponds and rivers. Most species are carnivorous and actively hunt for invertebrate prey. 

Most of the beetle families belong to the Polyphaga and they are listed alphabetically below. Buprestidae (14,480; 770) are called jewel beetles because of their coloured elytra. The larvae bore through roots, logs, stems, and leaves of various types of plants, ranging from trees to grasses. The subfamily Cicindelinae (tiger beetles) are known for their aggressive predatory habits and running speed. Cerambycidae (34,748; 6488) (longhorn beetles) feed on stems, trunks, or roots of both herbaceous and woody plants. Chrysomelidae (16,308; 2692) (leaf beetles) feed as larvae and adults on all sorts of plant tissue. Curculionidae (83,769; 6809) is the largest beetle family and its members are ‘true’ weevils or ‘snout beetles’ because of their distinctive long snouts. Lampyridae (2313; 155) (fireflies) are soft-bodied beetles, and conspicuously use bioluminescence during twilight to attract mates or prey. Light is produced from the lower abdomen by oxidation of ‘luciferin’ by the enzyme luciferase [3]. Fireflies are found in temperate and tropical climates, often in marshes or in wet, wooded areas where their larvae have abundant sources of food. Meloidae (873, 145) (blister beetles or oil beetles) occur primarily in temperate steppes and arid regions, as well as in subtropical and tropical savannas or other open habitats [4]. They are called blister beetles because of their defensive secretion of a blistering agent, cantharidin. Scarabaeidae (6770; 1161) (scarabs or scarab beetles) are stout-bodied beetles, many with bright metallic colours, with distinctive, clubbed antennae. The front legs of many species are adapted for digging. The C-shaped larvae, called grubs, are pale yellow or white, and live underground or under debris. Many scarabs are scavengers that recycle dung, carrion, or decaying plant material. Some are plant eaters. They have several subfamilies which are mentioned in this manuscript: 1. Cetoniinae (flower or fruit chafers); 2. Dynastinae (rhinoceros beetles); 3. Melolonthinae (May beetles); 4. Rutelinae (shining leaf chafers); and 5. Scarabaeinae (true dung beetles). Tenebrionidae (7241; 2505) (darkling beetles) are cosmopolitan in distribution and they occupy ecological niches mainly in deserts and forests as plant scavengers [5] (pp. 1151–1153). Most species are generalist omnivores, and feed on decaying leaves, rotting wood, fresh plant matter, dead insects, or fungi as larvae and adults.

After Section 2, I present the results of the interviews according to the different topics (insects as food, medicine, toys, etc.). In the discussion, arranged according to beetle families, the findings in the results are compared to what is known in the literature. 

## 2. Material and Methods

The information was collected by reviewing the literature and by personal interviews. The interviews were conducted in the years 1995 and 2000 in Africa and concentrated on the traditional, nutritional, and medical uses of arthropods and their products as well as on their role in religion, witchcraft, art, song, music, dance, children’s games, mythology, and literature. Although the information was collected some 20 years ago, it is probably all the more valid, because it would be difficult to obtain now. Due to rapid urbanisation and the fact that older people often had to be consulted by the interviewees, is a sign in itself that this type of indigenous knowledge is rapidly disappearing. 

Some of the results obtained in 1995 on insects have been published [6], as well as the part on edible insects over both years (1995 and 2000) [7].

The total number of people interviewed was 304 from 27 different countries in sub-Saharan Africa of whom 22 were resource persons (experts without recorded ethnic affiliation) (Table 1). The ethnic group was unknown for five other respondents. The total number of ethnic groups was 121, excluding Zanzibar and Madagascar, where the ethnicities were not recorded. Names of ethnic groups were checked, mostly in Wikipedia [8] and the Joshua project [9]. 

Most of the people interviewed were scientists or technicians trained in entomology. The interviewees were identified by visiting entomological groups of universities and (inter)national agricultural research institutes, plant protection services, museums, and crop protection projects. The author tried to interview most of the staff of these organisations (often arranged by the managers/directors in charge). The age of those interviewed varied between 25 and 65. Most of the respondents were male, reflecting the gender composition of the organisations. On a few occasions, people with no entomological background were interviewed in villages. This proved to be a challenge because of language and confusion about the insect species. Twenty-two of the respondents acted as resource persons on special topics (for example, experts on termites or insects as food or medicine) or had special positions (professors, heads of organisations, shamans, museum directors, and priests). In these cases, the ethnic origin of the person who provided the information was not considered relevant. 

The author used a list of issues to be covered in the interviews. Several respondents were sent this list before the country was visited. Often, they questioned elders, grandparents, family members and acquaintances before my arrival. This information was then passed on to me. In general respondents from rural areas were able to provide more information on the topics of interest than those from urban areas. 

Vernacular names and their meaning were double checked with the respondents and sometimes by literature search. The national libraries and university libraries in London and Paris, the library of the African Studies Centre in Leiden, the Netherlands, and some libraries of the countries visited were consulted. The literature consulted was mainly of an anthropological nature. Findings for a country or a certain tribe were only reported if information was received from more than one respondent, or if the information given during interviews was confirmed in the literature. The respondents’ countries and tribes are mentioned to specify the sources of information. They cannot be used for establishing correlations between ethnicity and information provided. The qualitative character of the information collected is emphasised.

## 3. Results

### 3.1. Food

Both the adults and the larvae of the Coleoptera are used as food. We will discuss water beetles, weevils, scarab beetles, jewel beetles and some other groups such as longhorn beetles.

#### 3.1.1. Water Beetles

Only one respondent of the CAR (Kari) mentioned that Gyrinidae are eaten. 

#### 3.1.2. Weevils

Almost all respondents from Cameroon, DRC, Benin, Nigeria, and Togo, mentioned the consumption of the palm weevil, *Rhynchophorus phoenicis* (Figure 1). The insect is mostly collected from the oil palm *Elaeis guineensis* and *Raphia* spp., but also from palm trees such as *Cocos nucifera*, *Metroxylon sagu*, and *Phoenix dactylifera*. After felling the palm, you must wait several days before you can harvest the larvae. The decision to harvest depends on the sound the larvae make in the trunk (Cameroon: Bafia, Bamileke, Yambassa; Togo: Akebu). The larvae can be found on rotten palms and on palms after the wine has been extracted. The adults can be eaten (removing the elytra) but it is most often the larvae that are consumed, after pulling the head off as well as removing the intestines; alternatively, they are pressed to remove the faeces, or the abdominal tip is cut off. The larvae are often barbecued (on a stick), roasted or boiled. They can also be cooked in soups and stews. From Mozambique (Makua) the cashew weevil stem borer *Mecocorinus loripes* is eaten. 

#### 3.1.3. Scarab Beetles

Cetoniinae. In Cameroon (Bamileke, Wimboum) they eat the adult (after removing the elytra) and in Chad (Ngambaye) also the larvae. 

Dynastinae. From the rhinoceros beetles, *Oryctes* spp. are consumed (Benin: Goun, Nagot; Cameroon: Bamileke, Bolous; DRC: Teke; Mbochi; Nigeria: Yoruba; CAR: Gbaya), and more specifically *O. monoceros* (Nigeria: Yoruba; Uganda: Ganda) and *O. gigas* in Madagascar. Both the adults and the larvae are eaten, but mostly the larvae. In Nigeria (Yoruba), before consumption they remove the elytra from the adults, called “Akake”, and from the larvae, called “Ogongo”, often collected from garbage heaps, they remove the bottom part of the abdomen and the intestines. Adults are a major pest of the coconut palm (*Cocos nucifera*) and the oil palm (*Elaeis guineensis*). In Cameroon (Bamileke; Bolous) they also consume *Augosoma* spp., pests of palms, coconut, and eucalyptus. 

Melolonthinae. *Eulepida mashoni* is eaten in Zimbabwe (Ndebele, Shona). The branches of *Julbernadia globiflora* and *Brachystegia speciformis* are shaken to collect the beetle. After removing the legs, they are washed, grilled and eaten as a relish. In Madagascar, *Hoplochelus marginalis* is eaten (fried in oil) both as adult (removing the legs and wings) and larvae. 

Scarabaeinae. The consumption of dung beetles was only mentioned once (Nigeria: Yoruba).

#### 3.1.4. Jewel Beetles 

People eat both the larvae (Zimbabwe: RES) and the adults (Burkina Faso: Mossi; Chad: Ngambaye; Mozambique: Bitonga, Tsonga-Rhonga, Tsonga-Tswa), after removing the elytra (Zambia: Tonga). The eggs are eaten in Mali (Sarakolé), collected by putting a female beetle in a box. *Sternocera orissa* (Figure 2) is eaten in southern Africa (Mozambique: Tsonga-Tswa; Zambia: Tonga). 

#### 3.1.5. Other Beetle Families

Cerambycid larvae are eaten in Benin (Tori) and Zambia (Namwanga, Tonga). 

### 3.2. Medicine

#### 3.2.1. Water Beetles

Adolescent girls in Cameroon (Bakossie, all nine respondents of the Bamileke, Bolous) collect adult water beetles and let them bite their nipples to stimulate breast growth. The same practice is used in Kenya (Kalenjin, Meru), Rwanda (Kiga-Toro); Uganda (Bunyoro, Ganda), and Zimbabwe (Shona, Zezuru). 

Other effects of the water beetles are to make you a better swimmer: by eating them (Zimbabwe: Shona), or letting them bite the tongue (Zambia: Tonga, RES) or the belly (Madagascar). The bite of the beetle in the tongue also makes you a better whistler (Burundi: Hutu; Zimbabwe: Ndebele, Zezuru). Several water beetles from the family Gyrinidae were mentioned for this purpose: *Gyrinus* spp., *Dineutus* spp., and *Aulonogyrus* spp. 

In Madagascar they cook and then eat Dytiscidae to prevent a cough.

#### 3.2.2. Blister Beetles

Meloids occur on crops but cause blisters, which is why people avoid them. That they produce blisters upon contact was frequently mentioned (Burkina Faso: Mossi; Chad: Mbaye; Mali: Mande-Mandinka; Niger: Songhay; Senegal: Diola, Fula, Serer, Wolof; Zambia: Tonga). The blisters can be treated using cowpea leaves (Niger: Hausa; Togo: RES) or the hairs of a hare or rabbit (Chad: Sara-Niellim). The blister beetles can also be used to kill someone by putting the powder or a liquid extract of the insect in food (Burkina Faso: Mossi; Chad: Ngambaye; Mali: Mande-Malinke, Mande-Mandinka, Tuareg; Niger: Hausa; Togo: Cotocoli, Ewe). For example, when two spouses of a husband have relationship difficulties, they may employ this method to one another (Senegal: Wolof). A preparation of the beetle is also used as an aphrodisiac (Mali: Mande-Malinke; Senegal: Wolof; Togo: RES; Zimbabwe: Shona), or to treat venereal diseases (Mali: Mande-Malinke; Zimbabwe: Zezuru), sometimes by mixing it with the roots of certain plants (Chad: Sara-Kaba, Sara-Niellim). Blister beetles are often used as medicine (Mali: Mande-Malinke;) by a medicine man or witchdoctors (Benin: Fon; Mali: Tuareg; Zambia: Nyanja; Zimbabwe: Shona). Powder of the insect is used to treat otitis (Senegal, Bainuk), warts (Zimbabwe: Shona) or in a drink to facilitate urination by children (Chad: Ngambaye). 

### 3.3. Hunting and Fishing

In South Africa respondents told me that the San use the pupae *Diamphidia nigroornata* (Chrysomelidae) (Figure 3) as an arrow poison. To strengthen the poison, they mix it with certain plant extracts: *Solanum coccineum* (Solanaceae), *Antizoma angustifalia* (Menispermaceae), *Boscia albitrunca* (Capparaceae), and *Acanthosycios naudinianus* (Cucurbitaceae). The beetle feeds on the small deciduous tree *Commiphora angolenis* (Burseraceae). The pupae of the chrysomelid are dug from the base of this tree. 

In Tanzania (Mwarusha) the larvae of Melolonthinae and Tenebrionidae are used for fishing. 

### 3.4. Toys

#### 3.4.1. Fireflies

Children play with fireflies, e.g., by trying to catch them (Cameroun: Bakoko, Bamileke; Chad: Goulaye, Ngambaye; Kenya: Luo; Mali: Sarakolé, Senufo; Niger: Songhai; Nigeria: Yoruba; CAR: Kari; Senegal: Serer, Wolof; Tanzania: Chaga, Iramba; Uganda: Ganda; Togo: Ewe; Zanzibar; Zambia: Bemba, Tonga). Children may put them in a bottle (Benin: Goun; Nigeria: Yoruba; Mozambique: Shona; Sudan: REF), tie a leg to a string (Nigeria: Yoruba), or crush them so that the glow remains in their hands (Benin: Fon; Gambia: Jola; Uganda: Bunyoro). However, the glowing insects also scare children (Nigeria: Yoruba; Togo: Kabye; Zanzibar).

#### 3.4.2. Dung Beetle

Children watch dung beetles rolling the dung ball (Sudan: Mahas; Tanzania: Chaga; Zambia: Tonga). They may consider them like buffalos (Madagascar). They play with them by placing the dung ball somewhere else and seeing whether the beetle can find it again (Burkina Faso: Mossi; Tanzania: Iraqw). They may put an enclosure around it and see whether the beetle can push the ball over it (Zimbabwe: Shona). They may also put two beetles together and let them fight (Tanzania: Mwarusha) in an arena (Mozambique: Shona). The dung balls are also used for throwing, playing soccer (Mozambique: Nchope, Zimbabwe: Tonga) or catapulting (Zimbabwe: Shona).

#### 3.4.3. Tiger Beetles

The larvae of the tiger beetle (Cicindelinae) live in vertical or slanting cylindrical burrows often a foot or more deep, in which they can move up and down with the aid of the dorsal hooks of the 5th abdominal segment. They are predacious and found along the sandy banks of rivers and bodies of water, in wet meadows and in damp partially shaded canyons [10]. In Kenya and Uganda (Luo) and in Madagascar children (boys and girls) play with these insects in the following manner. They insert a piece of grass in the tunnel. When the grass moves, they pull very fast and may catch the larvae and get it out (it has a long abdomen). It is used as a kind of competition to show how many larvae they can get out in this manner. In Madagascar, the beetles are also used for fighting matches.

#### 3.4.4. Beetles in General

In sub-Saharan Africa children commonly play with beetles in the following way. They let them fly in circles either by holding one end of a string which is attached to the leg of the beetle or by securing the beetle by tying it with a loop between the between the mesosoma (prothorax) and the metasoma (pterothorax + abdomen) (when tying it to the leg it may break off). However, a crueller method is to remove the tarsus and insert an acacia thorn in the tibia through which a string is attached which is connected to a stick (Burkina Faso: Mossi). In Sudan (Mahas, Rubatab), the method was slightly different. The tarsus and tibia are removed, and one claw of the two-claw acacia thorn was inserted in the hollow femur. Then they put a straw in the other claw rod and let the insect fly around. It makes a funny buzzing sound. Another method used is to insert a bamboo stick (Cameroon: Matha) or the nerf of a palm (50 cm) in the femur (Cameroon: Beti-Ewondo).

The use of beetles as toy aeroplanes was common for scarab beetles, most often Cetoniinae, in which the sound of the flight was very striking (Benin: Fon, Goun, Nagot, Tori; Cameroon: Bafia, Bamileke, Banen, Bani-Pahuin, Bassas, Beti-Ewondo, Matha, Wimboum; Chad: Ngambaye; Nigeria: Yoruba; Tanzania: Chaga, Iraqw, Mwarusha, Digo; Togo: Ewe, Mina). From the scarab beetles, species of different subfamilies were also mentioned: Dynastinae, *Oryctes* spp. (Nigeria; Yoruba); Melolonthinae (Madagascar); Scarabaeinae (dung beetles) (Tanzania: Pare). However, specimens from other beetle groups were also used, such as buprestids (Mali: Sarakolé; Mozambique: Bitonga, Tsonga-Rhonga, Tsonga-Tswa), viz. the three species: a green *Psiloptera* sp., a red-brown large *Sternocera castanea* and a smaller green one, *Steraspis speciosa*. Children had to climb the trees to get them. *Rhabdotis subrina* is used for the same purpose in the Kordofan region. The curculionid *Rhynchophorus* sp. was also mentioned (Cameroon: Bakoko, Bakossie, Beti-Ewondo; Nigeria; Yoruba; Togo: Akebu, Kabye), as well as big beetles in general (Burkina Faso: Mossi; Cameroon: Tikar; Madagascar; Mali: Songhay; Uganda: Luo; Sudan: Dongolawi, Fula, Nubian; Tanzania: Chaga; Zimbabwe: Shona). 

### 3.5. Religion and Superstition

#### 3.5.1. Fireflies

Fireflies were often associated with ghosts and witches (Benin: Fon, Goun, Nagot; Burkina Faso: Mossi; Cameroon: Bamileke, Banen, Bani-Pahuin, Beti-Ewondo, Bolous; Chad: Sara-Kaba; Sara-Niellim; DRC: Mbochi, Teke; Niger: Hausa, Songhai; Nigeria: Ebibio; CAR: Gbaya; Senegal: Wolof; Sudan: Kuku; Togo: Cotocoli, Ewe, RES; Uganda: Ganda, Nyoro; Zambia: Lozi; Zimbabwe: Zezuru; Zanzibar, Pemba). In Benin (Goun) a recipe with fireflies is used by a fetishist to liberate a person from a witch. From Mali (Mande-Mandinka) I was told that fetishists have light coming from their mouth, supposedly from fireflies. In Chad (Ngambaye) there is saying that a wizard shines like a firefly, but the wizard itself is not associated with fireflies. In Benin (Fon, Nagot) I was told that when you take an ointment of pigeon, a plant and fireflies and someone tries to poison you, the alcoholic drink will light up. In Kenya (Kikuyu) it was indicated that the light was given to the insect by God. In Zanzibar and Zambia (Lovale) it was believed that when you catch a firefly, you may get an inflammation under the nails. 

Four of the Luo respondents from Kenya told stories about night runners, which went as follows. Night runners (naked or wearing flimsy clothing) throw stones at doors and windows in order not to harm but to harass and scare people during the night. They do not want to be recognised. They can command snakes, leopards, lions, crocodiles, hippos, and other dangerous animals. Fire is often seen in their mouth and this is believed to be from fireflies (if you crush fireflies the glow will remain in your hand). When fireflies gather, a night runner is very likely to be present. Night runners may crush fireflies in their hands to scare people. The ability of a night runner is inherited and when you marry into a family of night runners you must adopt the same customs. 

In Benin (Bariba, Tori) and Cameroon (Bassas, Bani-Pahuin; Uganda: Bunyoro, Luo, Ganda) when fireflies enter the house it is not a good sign. However, others in Cameroon (Bafia, also Leiderer [11], Bassas, Bani-Pahuin, Beti-Eton, Matha, Yambassa; Senegal (Diola), Sudan (Fula) and Tanzania (Iraqw, Zanaki)) indicated that their arrival in the house means that something nice is going to happen or that a visitor is coming. Their presence is a sign of the end of the rainy season in Benin (Tori), Chad (Ngambaye), Mali (Sarakolé) and Zanzibar. In Zambia (Tonga), if you put them under a pot with water, it will rain the next day.

#### 3.5.2. Dung Beetles

Dung beetles are often considered dirty (Mozambique: Makua), and something you should not touch (Senegal: Serer), so children are often discouraged from handling them (Chad: Mbaye; Senegal: Diola). The saying ‘you are a dung beetle’ means you are a scumbag (Benin: Fon; Mozambique: Tsonga-Shangana; Zambia: Tonga).

The dung beetle is often associated with witchcraft (Benin: Nagot; Gambia: Jola, Mandinka; Senegal: Bainuk; Zambia: Tonga) or with bad luck, disease, or death (Burkina Faso: Mossi; Mali: Mande-Mandinka, Tuareg; Niger: Songhay; Senegal: Diola). However, it is also used as a charm to protect someone from evil (Sudan: Gaälien; Zambia: Nyanja, Namwanga). Dung beetles can find dung from a great distance. That is why it can be ground together with the roots of a certain tree and inserted in the nose of a hunting dog to increase their capacity to detect game (Zambia: Tonga)

#### 3.5.3. Flower Chafers 

In Mali when flower chafers (Cetoniinae) (Figure 4) arrive during the night, everybody flees, because they bring evil. When this insect touches somebody, they may become ill or even die and only the witchdoctor can reverse the curse (Chad: Goulaye; Mali: Mande-Malinke, Sarakolé; Senegal: Wolof). 

#### 3.5.4. Darkling Beetles

A tenebrionid pretends to be dead when disturbed (Zambia: Bemba, Namwanga, Tonga). It uses the characteristic ‘headstand’. Therefore, the beetle is used in witchcraft and as a protection against somebody who wants to attack you. Therefore, in Zimbabwe (Shona) mothers put the insect in their child’s hands and close it, keeping her hand over that of the child. They must hold this pose for some time before releasing the insect. In this way the child will not attract strangers. In the same country (Tonga) it was similarly used to prevent the child from taking food from others.

#### 3.5.5. Other Beetle Species

In Madagascar, the curculionid *Holonychus* sp. (likely *H. saxosus*) is wrapped in a piece of cloth and worn as a necklace for protection. This insect is very robust and solid and can withstand all kinds of problems. In Mali (Senufo), the appearance of *Acanthophorus confinus* (Cerambycidae) is a sign that that there will be a good millet harvest. It is captured and put in the grain storage. 

### 3.6. Nomenclature

In Togo (Ewe) the word for the water beetle is Zrotsi (Zro—purify; tsi—eau), so it is believed to purify water. Ten of my respondents from Cameroon (Bamileke, Bassas) mentioned the word “Lelê chieve” or “Lelochie” which means clearing (stagnant) water. The insects appear after the water has been disturbed, e.g., when mud appears after a rain shower. When the water is clean, they see them less. Children seem to put these insects in cloudy water to clear it. A respondent from Tanzania (Chaga) told me that Dytiscidae only swim in clear water, which is why his mother asks him to bring water from a pond where these insects are seen. When there are water beetles, it is a sign that there will be fish (Burundi: Hutu). 

The dung beetle in Yoruba (Nigeria) is called yimiyimi (“yi” is roll; “imi” is faeces); or in Pemba (Zanzibar) “Tuta mavi” (“tuta” is to take something from one place to another; “mavi” is dung); or in Mali (Fula) “Moussa”, i.e., roller of the cow ball.

In Cameroun fireflies they are called “fire of God” (Bamileke) or “bush lamp” (Wimboum).

### 3.7. Proverbs and Stories

In Sudan (Dongolawi, Kuku, Mahas; RES) there is a saying “*you eat like a grain weevil, but you do not grow fat*”, meaning that somebody eats a lot but does not put on any weight (beetles in storage always maintain the same weight). 

In Uganda (Acholi) the following story was told. An anteater wanted to marry a beautiful woman who was in heaven, but how could it get there? The firefly said: “*Don’t worry I shall give light, so we can find our way*”. The spider said: “*Don’t worry I shall make a web, so we have a path to take*”. The antlion said: “W*ell I can make a hole in the clouds so we can reach heaven*.” As the spider and the firefly had played a role in the enterprise, they got into an argument about who could marry the woman. Therefore, the anteater burrowed a hole, invited the spider and the firefly into it, filled in the hole, and married the woman. The meaning behind the story is that when you want to help a friend, you should not boast too much.

From Sudan (Fula, Gaälien, Kuku, Mahas, Rubatab, REF) the following story was often told. The dung beetle fell in love with the moon and asked her to marry him. The moon would only come down if the dung beetle cleared all the dung from the earth. Somebody remarked that in the story the beetle is a male while in fact it is a female. According to one person from this country (RES) it concerned *Heliocopris* spp. 

From Sudan (Fula), there is a story that after the rainy season the beetle collected a lot of material for the dry season. When the rainy season came, the frog begged for food. Then the dung beetle said: “*You have been enjoying yourself during the rainy season, singing instead of collecting food*”. Therefore, the dung beetle did not give any food because the frog had been enjoying himself instead of working during the rainy season.

From Kenya (Kamba) when you remove certain dung beetles from the manure and expose the insects to the sun, they change colour (become dark) in minutes, and they will die. Machakos is a dry district. Therefore, when somebody does not want to work because he is suffering in the sun, one could say “*Are you the larvae of the dung beetle?*” (you may change colour and die).

### 3.8. Art, Music and Commerce

The jewel beetle (Buprestidae) is known for its shining elytra. In Zambia (Tonga) they make necklaces from the elytra by threading them on a string. For the initiation ceremonies of a girl (preparation for womanhood) when she is 16/17 years old (preparation for marriage) she spends 1–3 months in isolated confinement. While isolated (not meant for an audience) she can make music by moving a string with the elytra of buprestids. In Kenya (Kamba) a necklace was made from elytra of the buprestid *Sternocera hildebrandti*, to be worn by women of a certain age of the Masai and Samburu. The elytra are alternated with seeds from a special legume, which are not eaten but specially used as an ornament. The beetles are caught from trees and brought to old women who make the necklace. It is only worn during special ceremonies because the elytra break easily. The elytra can also be used as a brooch (Madagascar).

Scarab beetles are also used in art and music. In Zambia (Tonga), the elytra from coloured flower chafer beetles are put on a string. When a girl is initiated, she is separated from the rest of the people and initiated in a whole series of things (how to look after her body, how to use her skills). Music is made from the rattling sound of these elytra. Additionally, in the Sudan (RES) scarabs were worn as necklaces, and it was mentioned that the adoration of scarabs came from Egypt. 

In Cameroon (Bamileke) children put a fruit chafer (Cetoniinae) in a match box and listen to the sound like a radio. 

In Nigeria (Yoruba) there seems to be an expression “when you see a beetle dancing on the water there must be drummers under the surface”, meaning when somebody behaves strangely there must be a reason for it. In Cameroon (Bani-Pahuin) the name of the beetle means “balaphone of the water”.

Due to their beauty, beetles may also be sold. For example, one of the largest beetles (5–11 cm) (*Goliathus goliatus*) is sold for between USD 100 and 3000. The jewel beetles also go for sums between USD 20 and 600. One beetle endemic to Madagascar is called the giraffe weevil, *Trachelophorus giraffa* (Curculionidae), and derives its name from its extended neck; they sell for about USD 10. 

## 4. Discussion

One of the questions in cultural entomology is how traditional knowledge relates to scientific knowledge. Western science aspires to be objective and quantitative as opposed to traditional knowledge, which is mainly subjective and qualitative [12]. Traditional knowledge embodies a wealth of wisdom and experience of nature gathered over millennia from direct observations and transmitted, most often orally, over generations. This knowledge can be spiritual in nature and does not make a distinction between empirical and sacred. 

Additionally, the distinction between medicine and superstition is hard to pin down. The term “similia similibus curantur” is appropriate; it literally means “likes are cured by likes” (the signature doctrine). In other words, either the appearance or the behaviour of the insect is believed to cure when employed. This appears in some of the findings presented. There are articles that deal with the cultural significance of beetles around the world, e.g., Clifford [13], but when it concerns Africa reference is often only made to the scarab beetle of Egypt. Similar studies encompassing the whole world would be a tremendous effort. Therefore, this article is restricted to the cultural importance of beetles in sub-Saharan Africa. The beetle families are mentioned in alphabetic order.

### 4.1. Buprestidae (Jewel Beetles)

It was shown in the results that in Mali, buprestid eggs (laid in a box by a female) are consumed. *Sternocera interrupta* eggs were also consumed in Cameroon [14], while the eggs of this species were extracted from the insect body in Botswana [15]. *Sternocera orissa* (Figure 2) is eaten in Botswana [15,16] and southern Africa [17] (p. 143), and collected from the trees *Acacia mellifera* and *Kyllinga alba* by the San [18]. In South Africa, *S. orissa* was prepared by roasting the insect in hot ash followed by removal of the wings and heads prior to eating [15,19]. In Zimbabwe, *S. orissa* are fried in large pans and later dried before they are sold [16]. *Sterapsis castanea* and *S. interrupta* are collected from acacia trees and consumed in Benin [20]. 

Jewel beetles (buprestids) have spectacular colours, which is why they are also used as ornaments, e.g., a necklace made with buprestid elytra from Tanzania [21]. However, also other beetle species were also used, such as a the curculionid *Brachycerus apteras* used to decorate a snuffbox from Zimbabwe (same author).

Muafor and Le Gall [22] describe the use of beetles (forest insects) as an income source in the southwest region of Cameroon and the contribution of this activity to improving rural livelihoods. Interviews conducted in 96 households in six villages revealed that the beetle trade provides a complementary household income to forest-dependent populations. They recommend formalising the insect trade, so that the resources can be managed sustainably. 

### 4.2. Carabidae (Ground Beetles)

I did not come across the eating of ground beetles, but in Madagascar the larvae of *Scarytes* sp. are consumed [23]. 

The stitching of wounds with ants is known in sub-Saharan Africa [24] but not with the mandibles of beetles, although in Algeria it seems to have been practiced. It concerns the carabid beetle *Scarites* sp. [25].

### 4.3. Cerambycidae (Longhorn Beetles) 

As shown cerambycid larvae are eaten in Benin and Zambia. They are also consumed in other countries in Africa: Senegal and Gabon *Ancylonotus tribulus* (Bodenheimer [17] (p.186); Säo Tomé and Principe, *Macrotoma edulus*; Senegal, *Omacantha gigas* [26]; Madagascar, *Hopliderus aquilius* [23]; and in Zambia, several *Acanthophorus* spp. [27] (pp. 231–237). 

### 4.4. Chrysomelidae (Leaf Beetles)

As mentioned in the results, the pupae of the chrysomelid *D. nigroornata* (Figure 3) are used as an arrow poison by the San in South Africa. In addition, three predatory carabid beetles whose larvae parasitise the larvae, *Lebistina subcruciata*, *L. holubi*, and *L. peringueyi* are also used and are probably even more poisonous than the chrysomelid pupae [28]. 

### 4.5. Curculionidae (Weevils)

It was shown that the larvae of the palm weevil *R. phoenicis* are a very popular food. For example, almost 90% of the Akan in southern Ghana eat this insect [29] and in Cameroon it is eaten by all ethnic groups [30]. Deciding the time for harvesting by the nibbling sound of the larvae was already reported from Togo in 1947 by Chesquière [31]. In Benin the larvae are eaten year-round [20,30]. In Madagascar the larvae of the palm weevils *Rhina* sp. and *Eugnoristus monachus* (Curculionidae) are consumed [23]. 

From the north western part of Cameroon the proverb “The palm beetle when eating the core of the palm tree does not know that by the time the palm dies, it too (the beetle) will die” [32] (p. 4). This refers to the fact that when extracting the larvae from the palm, the palm is decaying, and it means that the evil plan you have for others acts like a boomerang. From the same book (p. 13) there is a proverb “Do not be the bride who swallowed the beetle”, which means: do not refuse to do something in public, and then do it anyway in private. The story goes that a new bride was given some beetle soup but refused to eat it, although she had an appetite for it. During the night she stealthily swallowed it, but the beetles were not yet prepared and were still alive. The beetles gnawed at her bowels and she died. 

### 4.6. Dytiscidae and Gyrinidae (Water Beetles)

It was shown in the results that water beetles were associated with clean water and were even believed to purify water. Aquatic insects respond to a variety of environmental conditions and are increasingly used as indicators of water quality [33].

Although the eating of water beetles was mentioned once from Cameroon and Madagascar, they are used as human food as reviewed by Williams and Williams [34]. Dytiscidae (*Cybister* sp.) are eaten in Benin [20]; Cameroon [14]; Senegal, Sierra Leone and DRC [35]; Madagascar [23]; Togo [36]. 

Adolescent girls get water beetles to bite in their nipples to stimulate breast growth. Is this because the water beetle itself looks like a breast or is it because the defensive substance contains hormone-like steroids [37]? When the beetles ‘bite’ in a defensive reaction, they simultaneously secrete defensive substances (produced in special glands) to discourage possible vertebrate predators. Gyrinids produce, among other substances, norsesquiterpenes. The Dytiscidae also possess prothoracic defensive glands, which produce, among other substances, hormone-like steroids. Larvae of antlions (Neuroptera Myrmeleontidae) are similarly used in East Africa. This shows that one must be careful not to jump too quickly to conclusions about certain, seemingly illogical, practices.

In Madagascar there is a proverb indicating that a water beetle (Dytiscidae) gets fat on its own, meaning that one is responsible for one’s wellbeing. This is probably because they swim in clear water, and it is not clear where they get their food from [38] (p. 120). From the same author [38] (p. 81) and also about the same water beetle the following saying is mentioned: “*In spring (dry season) the beetle flies and is eaten by a bird, and when it comes on land, it is eaten by a lizard*”, so better stick to your own environment.

### 4.7. Lampyridae (Fireflies)

In sub-Saharan Africa, the glowing of fireflies has been associated with ghosts and witches as shown in the results. Iorkighir [39] (p. 105) when discussing witchcraft, reported that when a witch is asleep, the soul leaves his/her body. In this way the witch often flies to witch-meetings on the back of a strange animal like, for instance a firefly. The ethnic groups of Adja, Ga-Adangbe, Mina in Togo and Benin call a firefly “Adze”, which is a perpetrator of witchcraft [40]. Adze originated as disease-warning entities, specifically regarding the effects of mosquitos and malaria. If captured, their powers can be controlled, but once liberated from their human shells they take the form of miniscule fireflies with anteater-like trunks for sucking blood, and a firefly body. 

As shown in the results, among the Luo in Kenya night runners use fireflies. Kwanya [41] corroborated the stories about night runners, although the fireflies were not mentioned. The author concluded that night-running is a mysterious activity that many people do not understand and whose cultural and socioeconomic value should be investigated, debated, validated, and advocated. The article also mentioned that witches use the guise of harmless night-running to hurt, curse, or even kill their neighbours.

When an infant is having nightmares, the spirits can be chased away by preparing a drink from mixed crushed fireflies and ashes from the cooking fire where the infant resides [42]. Diluted in water the infant drinks it and it is also smeared on the forehead, head, ears, and nape. Fireflies are vanquishers of sorcerers and ashes signify the invisibility of the infant.

### 4.8. Meloidae (Blister Beetles)

In the results blister beetles are named in relation to the possibility of their use as a medicine for venereal diseases or to poison somebody. A traditional healer, n’anga of the Shona, proposed *Mylabris alternata* mixed with the roots of *Carissa edulis* as a remedy for venereal disease [19] (p. 308). However, they cautioned that it may result in death. The use of cantharidin as an aphrodisiac is known all over the world [43] and it also functions as a diuretic, and for the treatment of warts; again it comes with a warning that the consumption can be fatal [44]. Modern medicine has shown that cantharidin extracted from male blister beetles can be used as a medicine, e.g., it can suppress liver, pancreatic, colon, bladder and breast cancer [45]. 

### 4.9. Scarabaeidae (Scarab Beetles)

In the results it was shown that scarab beetles of the different subfamilies mentioned, except the dung beetles, are consumed in sub-Saharan Africa. That corresponds with the literature findings below.

#### 4.9.1. Melolonthinae 

*Melolontha* spp. are very much appreciated as food in Cameroon [46] and Gelfand [19] confirms the consumption of *Eulepida mashoni*, and Decary [23] the consumption of *Hoplochelus marginalis* in Madagascar. 

#### 4.9.2. Cetoniinae

From the flower chafers, the larvae of the goliath beetles (*Goliathus* spp.), among the largest insects on earth, are considered a delicacy in the DRC [47]. Adults of *Chondrorrhina abbreviata*, *Gnathocera varians*, *G. impressa*, *Pachnoda cordata*, *P. vossi*, *Polybaphes* sp. and *Rhabdotis bouchardi*) are consumed in Benin [20] and *Pachnoda marginata* in Cameroon [14]. The Fang-Pangwe of Cameroon forbid the consumption of *A. centaurus* to the uninitiated [Tessman (1913) cited in Ratcliffe [48]]. The San of southern Africa eat rhinoceros beetles and *A. centaurus* to acquire the “special powers” associated with these large beetles (same author citing Reitter, 1961).

In West Africa, Mali people in particular were shown to be afraid of these beetles, especially during the night; the species is *Diplognatha gagates* [49] (Figure 4). However, encountering one of the following species *Rhabdotis sobrina*, *Pachnoda cordata*, or *Pachnoda marginata*, is a sign that a visitor is coming [49]. 

In the results, it was shown that letting beetles fly when attached to a string is a popular game among children. To see whose beetle will stay airborne longest was also reported by Pemberton [50]: viz. *P. marginata* from Benin and *P. ephippiata* from Kenya. The same author mentioned the practice from Kentucky, USA. In Madagascar the use of beetles for this purpose was reported by Decary [51] (p. 150).

#### 4.9.3. Dynastinae 

*Oryctes boas* is consumed in Nigeria [52] and *O. owariensis*, and *O. monoceros* in South Africa [17] (p. 186). *Augosoma centaurus* and *A. hippocrates* are eaten in eastern Cameroon and two thirds of the consumers prefer the adults while the rest prefer the larvae [53]. 

#### 4.9.4. Rutelinae

Adults of *Popillia femoralis* were sold as food in the Dschang region of Cameroon [Ratcliffe [48] citing Lisle (1945)]. 

#### 4.9.5. Scarabaeinae 

There are about 2000 dung beetle species in Africa [54]. It is well known that Egyptians worship the Scarab beetle as the symbol of sun, light, and regeneration [55,56,57]. However, the mythology is not reflected in the thinking about the dung beetle in sub-Saharan Africa (the reference was only made once in Sudan). The beetle is more associated with dirt because they deal with manure. However, they are also very much admired because of their hard work. In addition, the rolling of the dung ball also inspires children’s games. The dung beetles *Heliocopris japetus* and *Pachylomera femoralis* are used as fish bait in Zambia [58].

From Zambia (Tonga) there is also a story of the dung beetle [59]. A long time ago Dung Beetle and Butterfly were best friends. When the First Man and First Woman passed by, they admired the Butterfly, so the Dung Beetle said “*Am I so ugly?*” “Not at all” said Butterfly. “*It is only because you have nothing to attract their attention. Humans admire beauty and strength. If you were to become the strongest insect in the world, for instance, they would surely take notice of you*.” Dung Beetle decided she would try and went off and practised all kinds of exercises. After coming back, she showed Butterfly how strong she had become. Dung Beetle fashioned some huge balls of elephant’s dung. As Dung Beetle was doing this, First Man and First Woman passed by. They were both so amazed by Dung Beetle’s prowess, that they did not even notice beautiful Butterfly. Dung Beetle was pleased with all the attention paid to her and she has been pushing dung balls around ever since.

In Nigeria there was a female chief of the Igbo, named Ahebi Ugbabe, who was crowned in 1918 [60] (pp. 114–117). Someone of “Ahebi” stature does not walk but is being carried on a hammock by four hefty men to the sound of Okanga music. One of the verses, “Ivu ada anyi ga ada”, relates in song to her coming and going from court. In this verse the carrying of Ahebi is compared with the skill of the dung beetle, known to carry more than its weight in soil and manure. The song tells us that the weight a dung beetle carries does not weigh it down. This is also an allusion to its weight in responsibilities: dung beetles improve the soil and remove droppings, protecting livestock from pests and flies.

### 4.10. Tenebrionidae (Darkling Beetles)

I did not come across the eating of Tenebrionidae larvae, although worldwide the mealworms *Tenebrio molitor*, *Alphitobius diaperinus* and *Zophobas moria* are industrially reared for food and feed. This is probably because Western insect-rearing companies had experience with rearing these insect species as pet food. 

The peculiar defensive attitude of the darkling beetle (Tenebrionidae), “headstanding”, was mentioned in the results. The behaviour is explained by Tschinkel [61]. It occurs when the insect is disturbed and happens prior to giving off a defensive secretion (such as with *Eleodes* sp.). This behaviour also evolved in species which lack defensive glands. In Zambia it has led to the belief that the defence posture of the beetle can be transferred to children. The same “headstanding” was the impetus behind the creation story of native Americans in Cochiti, New Mexico, United States [62] (p. 35). This behaviour of the *Eleodes* beetle was explained as the beetle being ashamed (bending the head) for having dropped stars in the Milky Way, while forming the galaxy [63]. Therefore, here the headstand was interpreted in a different way.

### 4.11. Beetles in General

In Madagascar the larvae of *Prosopocoilus serricornis* (Lucanidae) are eaten [23]. 

A story from South Africa [64] (pp. 124–126) goes as follows. Lion, the king of all the animals, put on his fancy robes, his jewelled crown, and all his gold and silver medals. His clothes were very heavy, but they made him look mighty and grand. “*I am your king!*” he roared in delight at a meeting where many animals were present. There was also the tiny Beetle, who was so little that he had to walk on the side of the road so that the bigger animals would not step on him by accident. Then Lion saw the tiny Beetle, standing alone by the side of the road. The Beetle was singing to himself as he watched the Lion: “*I am small. I am little. But inside, I’m tall as a tree!*” Lion said to the Beetle: “*You, Beetle, bow before me!*” Beetle said: “*Your Royal Majesty, I know that I am small, but if you look at me closely, you will see that I am making a bow!*” Lion leaned over, peering down at the tiny beetle. His splendid robes, his jewelled crown and his many medals were so heavy that he lost his balance and he fell over and rolled into a ditch filled with muddy water. All the animals laughed. He was not so very mighty, after all.

## 5. Conclusions

Many beetle species are eaten, mainly from the families Curculionidae, Scarabaeidae, Buprestidae, Cerambycidae and water beetles of the family Dytiscidae. The hard elytra are often removed from the adults, while the intestines are stripped from the larvae, certainly in the case of *Oryctes* spp., which are often found in garbage heaps. In the tropical humid and subhumid eco-agroecological zones of sub-Saharan Africa the most popular edible insect species is the palm weevil *R. phoenicis*.

Some beetle species are used as medicine. Adolescent girls use dytiscids to stimulate breast growth. It is difficult to know whether the signature doctrine is applicable because the beetles are breast shaped or because the defensive secretion, containing steroid hormones, has a stimulating effect. Blister beetles (Meloidae) have an active substance cantharidin. They seem to influence the urinary tract, which is probably why they are used as an aphrodisiac and for treating venereal disease. In modern medicine cantharidin is used, among others, to treat cancer.

It is amazing that throughout sub-Saharan Africa children play similar games with beetles. The most popular activity is to let them fly attached to a string. It is not clear whether this also occurs in other parts of the world, although there was a reference from the USA. Other species children play with are dung beetles (when rolling the dung ball) or tiger beetles (extracting the larvae from their tunnels).

Throughout sub-Saharan Africa, fireflies are associated with ghosts and witches. This is understandable as the phenomenon of glowing beetles is spectacular and magical. The defensive attitude of the darkling beetle is used to protect the child. 

In the Sahelian part of Africa dung beetles are predominant. That is probably why in the Sudan there were several stories explaining the hard work of the dung beetles. 

Many beetle groups are brightly and attractively coloured, making them objects of collection and decorative displays. One family, the Buprestidae, even derived their name from an object of beauty, jewel beetles. However, other groups also display beautiful colours, such as for example the scarab beetles, the longhorn beetles, and the tiger beetles. They are used in ceremonies, jewels and commerce. 

This information was collected some 20 years ago and even then, interviewees often had to consult (grand)parents or elders to get the knowledge. People from rural areas provided more information than those from urban areas. This shows that with urbanisation and westernisation this kind of indigenous knowledge is rapidly disappearing. However, from all the information collected, it can be concluded that beetles play a major role in peoples’ lives, as food, toys, in stories, and in spiritual life.

## Figures and Tables

**Figure 1 insects-12-00368-f001:**
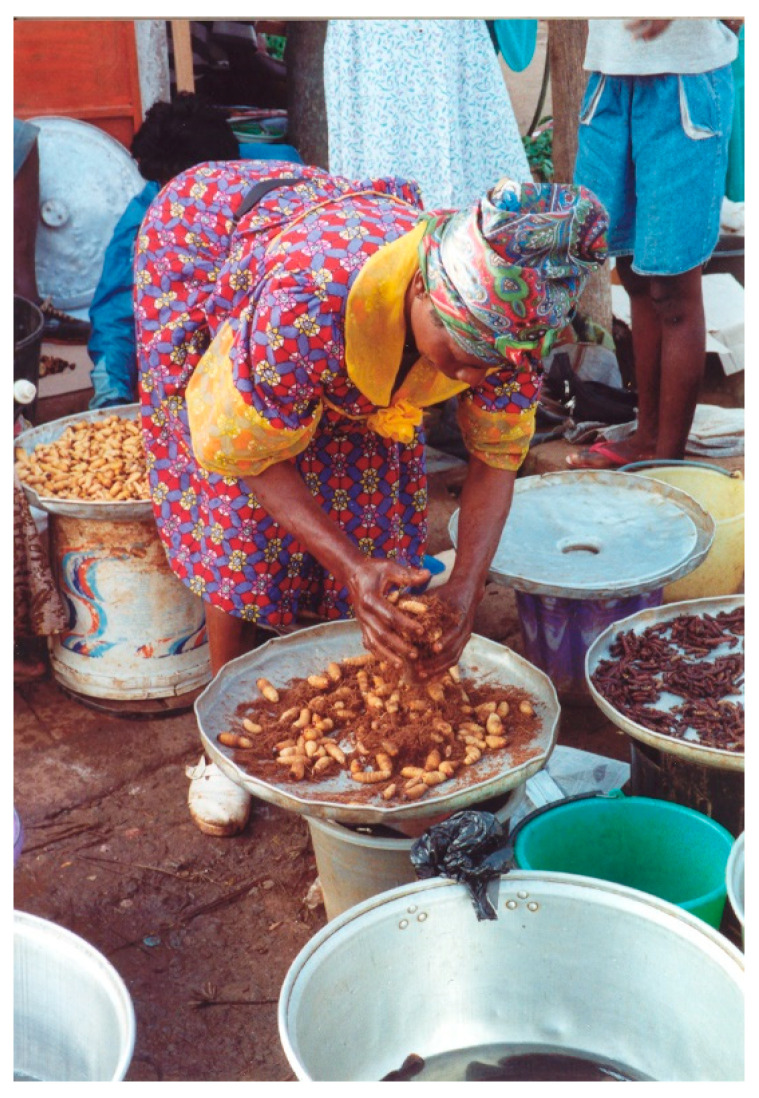
Selling the larvae of the African palm weevil *Rhynchophorus phoenicis* on the local market in Yaoundé, Cameroon (photo by author).

**Figure 2 insects-12-00368-f002:**
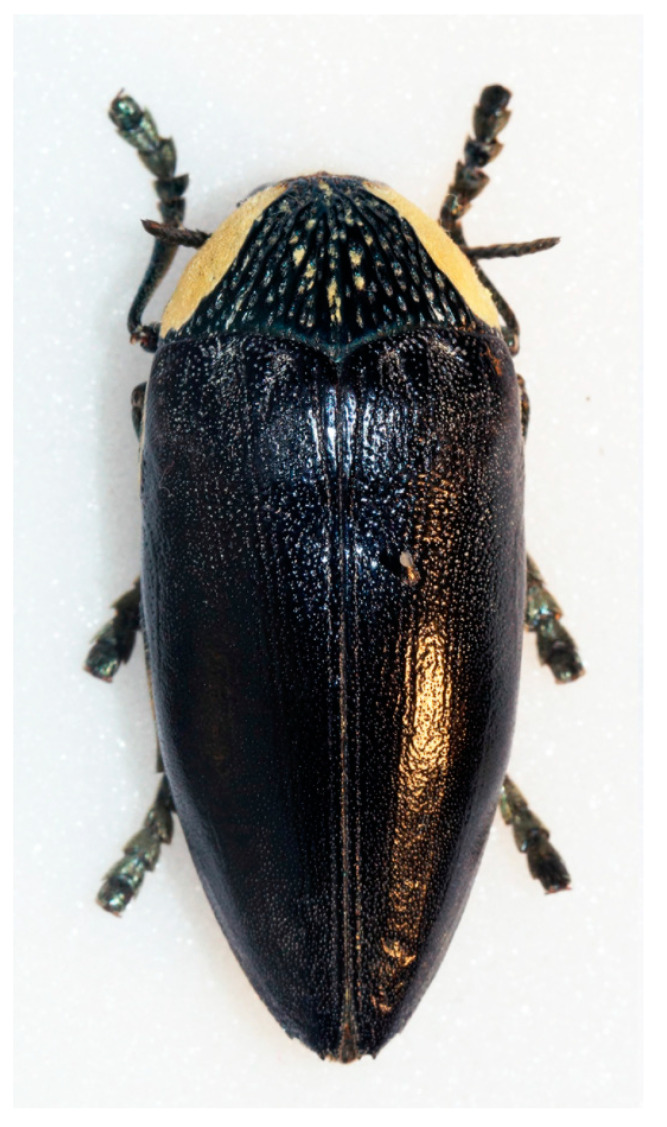
The jewel beetle *Sternocera orissa* (Buprestidae) is eaten in southern Africa. https://commons.wikimedia.org/wiki/File:Sternocera_orissa_monacha_-_Flickr_-_Bennyboymothman.jpg (Accessed on 20 April 2021). Attribution: Ben Sale from UK, CC BY 2.0, via Wikimedia Commons.

**Figure 3 insects-12-00368-f003:**
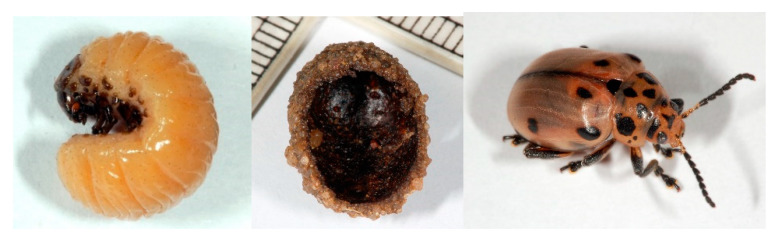
The larva, the pupa and the adult of *Diamphidia nigroornata* (Chrysomelidae). The pupae are used as arrow poison by the San in southern Africa. https://commons.wikimedia.org/wiki/File:Diamphidia_fg01.jpg (also fg03 and 04). Attribution: Fritz Geller-Grimm, CC BY-SA 2.5, via Wikimedia Commons. (accessed on 20 April 2021)

**Figure 4 insects-12-00368-f004:**
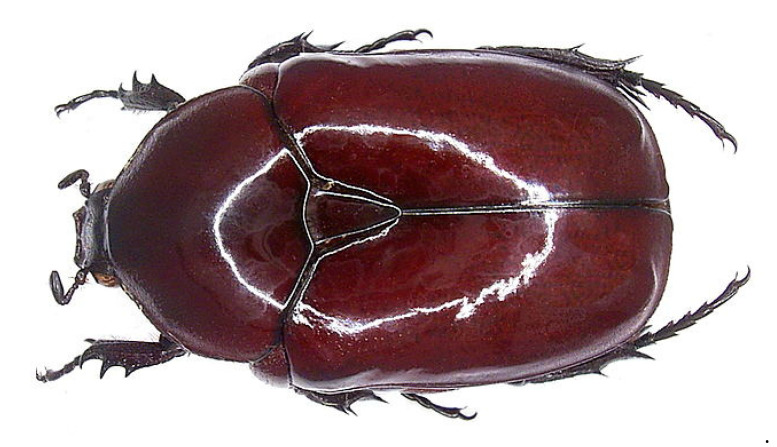
The feared scarab beetle *Diplognatha gagates* (Cetoniinae) in Mali. (https://commons.wikimedia.org/wiki/File:Diplognatha_gagates_Forster,_1771_(3232045262).jpg). Attribution: Udo Schmidt from Deutschland, CC BY-SA 2.0, via Wikimedia Commons. (Accessed 20 April 2021).

**Table 1 insects-12-00368-t001:** The number of respondents (N) per country and ethnic group.

Country	Ethnic Group—N	RES ^1^	N
Benin	Bariba—1, Fon—4, Goun—1, Nagot—6, Popo—1, Tori—1		14
Burundi	Hutu—2		2
Burkina Faso	Mossi—4, Fula—1		5
Cameroon	Bafia—1, Bakoko—1, Bakossie—1, Bamileke—14, Banen—1, Bani—Pahuin—1, Bassas—2, Beti-Eton—1, Beti-Ewondo—1, Bolous—1, Matha—1, Tikar—1, Wimboum—1, Yambassa—1	2	28
CAR ^2^	Gbaya—1, Kari—1		2
DRC ^3^	Mbochi—1, Teke—1		2
Chad	Arabe—1, Goulaye—2, Kanembou—1, Mbaye—2, Ngambaye—7, SaraKaba—1, SaraNiellim—1, Tupuri—1, Wadai—1		17
Gambia	Jola—1, Mandinka—1		2
Guinee-Bissau	Balanta—1		1
Kenya	Kalenjin—1, Kamba—4, Kikuyu—2, Luo—4, Meru—1, Somalian— 1		13
Madagascar	-		24
Malawi	Chewa—1		1
Mali	Fula—1, Mande-Malinke—1, Mande-Mandinka—1, Sarakolé—1, Senufo—2, Songhay—3, Tuareg—1		10
Mozambique	Bitonga—1, Makua—1, Nchope—1, Shona—1, Tsonga-Rhonga—2, Tsonga- Shangana—1, Tsonga-Tswa—1		8
Namibia	Damara—1		1
Niger	Djerma—1, Hausa—9, Kanuri—1, Songhai—4	1	15
Nigeria	Ebibio—1, Ebira—1, Yoruba—15, unknown—1		18
Rwanda	Kiga-Toro—1		1
Senegal	Bainuk—1, Diola—4, Fula—1, Halpulaar—2, Lebu—1, Serer—3, Wolof—5		17
South Africa	-	6	
Sudan	Dongolawi—1, Fula—1, Gaälien—3, Kambari-Abadi—1, Kawahla—1, Kuku—1, Mahas—1, Nubian—1, Nubian-Mahas—1, Rubatab—2, Tunyur—1, unknown—4	5	18
Tanzania	Chaga—7, Digo—1, Iraqw—3, Iramba—1, Mwarusha—2, Pare—1, Rangi—1, Sukuma—2, Zanaki—1	1	19
Togo	Akebu—1, Ewe—5, Cotocoli—1, Kabye—1, Mina—1	3	9
Uganda	Acholi—1, Banyankole—1, Bunyoro—1, Busoga—1, Ganda—7, Langi—1, Luo—2, Nyoro—1		15
Zambia	Bemba—1, Ila—1, Lovale—1, Lozi—2, Lunda—1, Namwanga—2, Nyanja-Chewa—1, Tonga—10, Tumbuka—1	2	20
Zanzibar			9
Zimbabwe	Ndebele—1, Shona—9, Zezuru—1	2	11
**Total number of resource persons and respondents**		**22**	**282**

^1^ Resource persons. ^2^ Central African Republic. ^3^ Democratic Republic of Congo.

## Data Availability

The data presented in this study are available on reasonable request from the author. The data are not publicly available due to privacy reasons.

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
