# Peer review of "Cultural Significance of Beetles in Sub-Saharan Africa"

_insects, 2021, doi:10.3390/insects12040368_

Round 1

Reviewer 1 Report

This is an interesting study based on interviews to more than 300 people from 27 countries. In cultural entomology and ethnozoology, many conventional studies are usually results of literature searching. Therefore, this paper is unique and appropriate for publishing on Insects. 

However, there are no intercultural comparisons in the discussion and conclusion. For example, author writes "In sub-Saharan Africa, the glowing of fireflies has been associated with ghost and witches " (514 line). In Southeast Asia and Japan, fireflies are looked upon as ghosts, too. If possible, author should comment similarity and heterogeneity of cultural significance of beetles between sub-Saharan Africa and other continents. 

Author Response

Query: This is an interesting study based on interviews to more than 300 people from 27 countries. In cultural entomology and ethnozoology, many conventional studies are usually results of literature searching. Therefore, this paper is unique and appropriate for publishing on Insects.

Answer: thanks

Query: However, there are no intercultural comparisons in the discussion and conclusion. For example, author writes "In sub-Saharan Africa, the glowing of fireflies has been associated with ghost and witches " (514 line). In Southeast Asia and Japan, fireflies are looked upon as ghosts, too. If possible, author should comment similarity and heterogeneity of cultural significance of beetles between sub-Saharan Africa and other continents.

Answer: I agree, but this would be a major job and enlarge the article tremendously. This was mentioned in the discussion (line 501-505).

Reviewer 2 Report

The manuscript describes how people view coleopteran taxa in the Afrotropical area. The topic could be interesting, although it is essentially a descriptive report, since studies on these topics a re scarce.

The author subdivided the sections by topics (as food, religion and so on), but it could be better first use the families (Carabidae, Chrysomelidae, and so on) and then show the different topics for each family. It could be more interesting for an entomologist, while the present subdivision would likely be more appealing for an ethnographer.

The manuscript could have some interest for an entomologist, but I suggest a more "entomological" than "anthropological" presentation.

Author Response

Query: The manuscript describes how people view coleopteran taxa in the Afrotropical area. The topic could be interesting, although it is essentially a descriptive report, since studies on these topics a re scarce.

Answer: There is very little information available. What is mentioned in the discussion is all the literature information I could find.

Query: The author subdivided the sections by topics (as food, religion and so on), but it could be better first use the families (Carabidae, Chrysomelidae, and so on) and then show the different topics for each family. It could be more interesting for an entomologist, while the present subdivision would likely be more appealing for an ethnographer.

Answer: The families are mentioned in the background information. I had to separate results from discussion. That is the journal’s format. I started writing ‘results and discussion’, which for this article may have been better. However, that was not possible. I decided to give the results according to topics and the discussion (in which the literature is mentioned) according to beetle families.

Query: The manuscript could have some interest for an entomologist, but I suggest a more "entomological" than "anthropological" presentation’.

Answer: Cultural entomology is the interface between "entomological" and "anthropological"

Reviewer 3 Report

The author has collected a remarkable treasure of information on oral traditions across a startling number of countries and ethnic groups. What makes this information so valuable is the fact that those traditions are being lost, some without record. There are few scholarly journals within which such information can be published, and I believe all of what is included should be published in Insects. What the author offers are vestiges of a dying knowledge base, akin to the global loss of languages. I fully support the publication of this article, but with minor reworking and changes, as suggested below. The most important critiques relate to organization of material, which is problematic as presented. Numbers refer to line numbers in the submission.

10-11: I suggest deleting the question related to breast growth, as no other listing in the Simple summary is longer than one sentence, or is accompanied by a question. Too much attention is paid to this topic relative to the others if the second sentence (the question) is included. It is fine as written in the Abstract.

15-16: As written, it is unclear if the information is being lost, or the practices (and, consequently, the information associated with the practices) are being lost.

20: Cerambycidae misspelled

27: Instead of "looks magical," I suggest "is interpreted as magical."

28-29: See comment for 15-16, above.

32: I found the background information of the beetle families to be refreshingly general and informative. Because all background information about the history (precedence) of cultural coleopterology is placed in the Discussion instead of in the Background section, I recommend alluding to this at the end of the Background section. I was looking for this essential information and was perplexed to find the only background material to be about beetle biology.

33: I recommend either adding "as of ____ [date of submission]" or adding date website was accessed under literature cited when referring to the very specific numbers of beetle species and genera.

50: Please cite reference for naming of Gyrinidae; I was under the impression that the name came from their (undisturbed) surface-water searches (and not due to movements associated with being alarmed).

52: Delete "any" unless there is evidence that they would actively hunt ants, centipedes, etc.

75: Dynastinae misspelled.

80: Replace "and" with "or" to be potentially (but not necessarily) inclusive.

106: Place comma after “On a few occasions”.

116-117: Please clarify. Do you mean that rural respondents knew more on the topics of interest, or received more information from the list supplied to them, or from some other source associated with the interview process?

126-127: What is meant by this final sentence? Is the author emphasizing the qualitative nature of the data presented here, or elsewhere?

128: A visual distinction needs to be made among thematic subheadings (e.g., Food) vs. insect groups used (e.g., Weevils) to make it easier for the reader. It is presently inconsistent, so confusing. This can be achieved by consistently using bold/italics with hierarchical numbering or lettering.

173: Replace semicolon with comma (because what follows the semicolon does not stand alone as a new sentence, but is a continuation of the same thought).

217: Please clarify if the two spouses would use the method of poisoning on each other or on the shared husband.

244-245: It seems as though some of this material should go in the "Nomenclature" and "Religion and superstition" sections.

254-256: I suggest not beginning sentences with "Or..."

272-273: The loop would have to be between the mesosoma (prothorax) and the metasoma (pterothorax + abdomen) in beetles, not the "abdomen and thorax."

292: Italicize genus name and spell correctly (Rhynchophorus).

310: Inconsistent capitalization of "Night" runners.

328-331: Does this really fall under this section? This seems more like a hygienic practice, but I suppose it has to do with how the sentiments are expressed.

336-338: This belongs under "Hunting and fishing."

363: I suggest reordering the subjects so that "Proverbs and stories" is next to "Nomenclature." Also, "Art and music" includes commerce (sale of beetles) rather than art. I would move this out of the Art and music section.

367-375: Be consistent spelling "anteater."

420: The Discussion has a wealth of information that is relevant and all of it should be included. Upon first reading, I found the organization of the paper a bit confusing because almost all of the Discussion material would conventionally be placed in the Introduction/background. As presented, the Discussion includes some points of comparison (with the author's findings), and these serve as more appropriate material for the typical Discussion section. I recommend preparing the reader for the taxon-grouped information with a sentence or two about how the information compares with past findings. A segue between line 434 and what follows would help.

422: "objective" should be "aspires/attempts to be objective"

425: "is spiritual" should be "can be spiritual"

476: groups (vs. group)

504: Myrmeleontidae (misspelled)

558: "people" repeated

573: Be consistent with font size.

600: "illusion" should be "allusion"

635: spp should be spp.

666-678: This section should be eliminated, with information included in the respective figure legends.

666-678: All species names should be italicized.

Author Response

Query: The author has collected a remarkable treasure of information on oral traditions across a startling number of countries and ethnic groups. What makes this information so valuable is the fact that those traditions are being lost, some without record. There are few scholarly journals within which such information can be published, and I believe all of what is included should be published in Insects. What the author offers are vestiges of a dying knowledge base, akin to the global loss of languages. I fully support the publication of this article, but with minor reworking and changes, as suggested below. The most important critiques relate to organization of material, which is problematic as presented. Numbers refer to line numbers in the submission.

Answer: Thanks, I did my best in meeting the concerns mentioned above. The line numbers did not always correspond with the article I downloaded from the Journal. Therefore, I mention those lines in (parantheses) behind the line mentioned by the reviewer.

Query 10-11: I suggest deleting the question related to breast growth, as no other listing in the Simple summary is longer than one sentence, or is accompanied by a question. Too much attention is paid to this topic relative to the others if the second sentence (the question) is included. It is fine as written in the Abstract.

Answer: Done, deleted.

15-16: As written, it is unclear if the information is being lost, or the practices (and, consequently, the information associated with the practices) are being lost.

Answer: If the information is lost, then with it the practices. But I changed it into “indigenous knowledge” being lost (also M&M, line 96, and in the conclusion, line 723)

Query 20: Cerambycidae misspelled

Answer: Corrected

Query 27: Instead of "looks magical," I suggest "is interpreted as magical."

Answer: Accepted

Query 28-29: See comment for 15-16, above.

Answer: I changed it into “indigenous knowledge” being lost (see above).

Query 32: I found the background information of the beetle families to be refreshingly general and informative. Because all background information about the history (precedence) of cultural coleopterology is placed in the Discussion instead of in the Background section, I recommend alluding to this at the end of the Background section. I was looking for this essential information and was perplexed to find the only background material to be about beetle biology.

Answer: I found it more logical to first give the information I collected and then afterwards how that relates to what was found in the literature in the discussion. I first had results and discussion together, but that is not allowed by the journal. As suggested, I have now mentioned this at the end of the background chapter to inform the reader about this set-up.

Query 33: I recommend either adding "as of ____ [date of submission]" or adding date website was accessed under literature cited when referring to the very specific numbers of beetle species and genera.

Answer: The date of accession is mentioned in the reference itself (15 March 2021)

Query 50: Please cite reference for naming of Gyrinidae; I was under the impression that the name came from their (undisturbed) surface-water searches (and not due to movements associated with being alarmed).

Answer: I have left away “when alarmed”. It was mentioned in Wikipedia and they referred to Imms, but in the latter I could not find it. Thanks for this.

Query 52: Delete "any" unless there is evidence that they would actively hunt ants, centipedes, etc.

Answer: Deleted

75: Dynastinae misspelled.

Answer: corrected

80 (86): Replace "and" with "or" to be potentially (but not necessarily) inclusive.

Answer: Accepted

106 (124): Place comma after “On a few occasions”.

Answer: Accepted

Query 116-117 (134-135): Please clarify. Do you mean that rural respondents knew more on the topics of interest, or received more information from the list supplied to them, or from some other source associated with the interview process?

Answer: Changed as follows: “In general respondents from rural areas were able to provide more information on the topics of interest than those from urban areas.” I hope that this is clear now.

Query 126-127 (144-145). What is meant by this final sentence? Is the author emphasizing the qualitative nature of the data presented here, or elsewhere?

Answer: I changed “information provided” into “information collected”.

Query 128 (148): A visual distinction needs to be made among thematic subheadings (e.g., Food) vs. insect groups used (e.g., Weevils) to make it easier for the reader. It is presently inconsistent, so confusing. This can be achieved by consistently using bold/italics with hierarchical numbering or lettering.

Answer: I agree, and the problem is that food was made into a heading 3 but should have been a heading 2. I changed it. I will alert the editors about the proper headings in the whole manuscript, because it went also wrong in other places.

Query 173 (209): Replace semicolon with comma (because what follows the semicolon does not stand alone as a new sentence, but is a continuation of the same thought).

Answer: I believe it is line 209. Done.

Query 217 (254): Please clarify if the two spouses would use the method of poisoning on each other or on the shared husband.

Answer: Added “to one another”

Query 244-245 (279): It seems as though some of this material should go in the "Nomenclature" and "Religion and superstition" sections.

Answer: Agreed. Moved to nomenclature section and to the religion and superstition section.

Query 254-256 (287): I suggest not beginning sentences with "Or..."

Answer: “Or” removed

272-273 (312-313): The loop would have to be between the mesosoma (prothorax) and the metasoma (pterothorax + abdomen) in beetles, not the "abdomen and thorax."

Answer: I was not aware about this morphology, but I have accepted it and changed it accordingly.

292 (332): Italicize genus name and spell correctly (Rhynchophorus).

Answer: done (also in figure 1 and line 153)

Query 310 (354): Inconsistent capitalization of "Night" runners.

Answer: Changed into night runners everywhere (no capital)

Query 328-331 (372-375): Does this really fall under this section? This seems more like a hygienic practice, but I suppose it has to do with how the sentiments are expressed.

Answer: I leave it as it is, as indeed it is about sentiments.

Query 336-338 (380-382): This belongs under "Hunting and fishing."

Answer: I understand. However, this looks more related to the doctrine of signature. Therefore. I leave it as it is.            

Query 363 (409): I suggest reordering the subjects so that "Proverbs and stories" is next to "Nomenclature." Also, "Art and music" includes commerce (sale of beetles) rather than art. I would move this out of the Art and music section.

Answer: Thanks for informing me. I have moved the chapter “Nomenclature” before the chapter “Proverbs and Stories”. I changed the title of “Art and music” into “Art, music and commerce”. I noticed that the Heading level for the last two was one (Heading 3), but these are Headings level 2. I changed it.

Query 367-375 (431-439): Be consistent spelling "anteater."

Answer: changed now everywhere in “anteater”

Query 420 (491): The Discussion has a wealth of information that is relevant and all of it should be included. Upon first reading, I found the organization of the paper a bit confusing because almost all of the Discussion material would conventionally be placed in the Introduction/background. As presented, the Discussion includes some points of comparison (with the author's findings), and these serve as more appropriate material for the typical Discussion section. I recommend preparing the reader for the taxon-grouped information with a sentence or two about how the information compares with past findings. A segue between line 434 and what follows would help.

Answer: I have tried to do so, but it may not have been clear. I have now added as reference “in the results in several places. I have also made at line 434 (497-498) a new paragraph.

Query 422 (492): "objective" should be "aspires/attempts to be objective"

Answer: Accepted “aspires to be objective”.

Query 425 (496): "is spiritual" should be "can be spiritual"

Answer: Accepted

Query 476 (549): groups (vs. group)

Answer: Accepted

Query 504 (577): Myrmeleontidae (misspelled)

Answer: Corrected

Query 558 (634): "people" repeated

Answer: ‘people’ (second time) removed.

Query 573 (650): Be consistent with font size.

Answer: That has been corrected. But I found that the four subfamilies of the  Scarabaeidae (scarab beetles) had also Heading 3. I have now changed it into Heading 4. Not sure how the editors will deal with it. The heading can also be left away. But it should be clear that it are subheadings of the  Scarabaeidae (scarab beetles).

Query 600 (677): "illusion" should be "allusion"

Answer: Correct. Done.

Query 635 (713): spp should be spp.

Answer: Done

Query 666-678 (754-759): This section should be eliminated, with information included in the respective figure legends.

Answer: I think this section was added by the editors.

Query 666-678 (754-759): All species names should be italicized.

Answer: Done

Round 2

Reviewer 1 Report

There are no problems in revised manuscript. 

Author Response

I do not see any comments by Reviewer 1,